# AUXILIARY TASK DISCOVERY THROUGH GENERATE-AND-TEST

## ABSTRACT

In this paper, we explore an approach to auxiliary task discovery in reinforcement learning based on ideas from representation learning. Auxiliary tasks tend to improve data efficiency by forcing the agent to learn auxiliary prediction and control objectives in addition to the main task of maximizing reward, and thus producing better representations. Typically these tasks are designed by people. Meta-learning offers a promising avenue for automatic task discovery; however, these methods are computationally expensive and challenging to tune in practice. In this paper, we explore a complementary approach to the auxiliary task discovery: continually generating new auxiliary tasks and preserving only those with high utility. We also introduce a new measure of auxiliary tasks' usefulness based on how useful the features induced by them are for the main task. Our discovery algorithm significantly outperforms random tasks, hand-designed tasks, and learning without auxiliary tasks across a suite of environments.

## 1 INTRODUCTION

The discovery question—what should an agent learn about—remains an open challenge for AI research. In the context of reinforcement learning, multiple components define the scope of what the agent is learning about. The agent's behavior defines its focus and attention in terms of data collection. Related exploration methods based on intrinsic rewards define what the agent chooses to do outside of reward maximization. Most directly, the auxiliary learning objectives we build in, including macro actions or options, models, and representation learning objectives force the agent to learn about other things beyond a reward maximizing policy. The primary question is where do these auxiliary learning objectives come from?

Classically, there are two approaches to defining auxiliary objectives that are the extremes of a spectrum of possibilities. The most common approach is for people to build the auxiliary objectives in pre-defining option policies, intrinsic rewards, and model learning objectives. Although most empirically successful, this approach has obvious limitations like feature engineering of old. At the other extreme is *end-to-end learning*. The idea is to build in as little inductive bias as possible including the inductive biases introduced by auxiliary learning objectives. Instead, we let the agent's neural network discover and adapt internal representations and algorithmic components (e.g., discovering objectives (Xu et al., 2020), update rules (Oh et al., 2020), and models (Silver et al., 2017)) just through trial and error interaction with the world. This approach remains challenging due to data efficiency concerns and in some cases shifts the difficulty from auxiliary objective design to loss function and curriculum design.

An alternative approach that exists somewhere between human design and end-to-end learning is to hand-design many tasks in the form of additional output heads on the network that must be optimized in addition to the primary learning signal. These tasks, called *auxiliary tasks*, exert pressure on the lower layers of the neural network during training, yielding agents that can learn faster (Mirowski et al., 2016; Shelhamer et al., 2016), produce better final performance (Jaderberg et al., 2016), and at times transfer to other related problems (Wang et al., 2022). This positive influence on neural network training is called the *auxiliary task effect* and is related to the emergence of good internal representations we seek in end-to-end learning. The major weakness of auxiliary task learning is its dependence on people. Relying on people for designing auxiliary tasks is not ideal because it is

challenging to know what auxiliary tasks will be useful in advance and, as we will show later, poorly specified auxiliary tasks can significantly slow learning.

There has been relatively little work on autonomously discovering auxiliary tasks. One approach is to use meta learning. Meta-learning methods are higher-level learning methods that adapt the parameters of the base learning system, such as step-sizes, through gradient descent (Xu et al., 2018). This approach can be applied to learning auxiliary tasks defined via General Value Functions or GVFs (Sutton et al., 2011) by adapting the parameters that define the goal (cumulant) and termination functions via gradient-descent (Veeriah et al., 2019). Generally speaking, these meta-learning approaches require large amounts of training data and are notoriously difficult to tune (Antoniou et al., 2018).

An exciting alternative is to augment these meta-learning approaches with generate-and-test mechanisms that can discover new auxiliary tasks, which can later be refined via meta-learning. This approach has produced promising results in representation learning where simple generate-and-test significantly improve classification and regression performance when combined with back-prop (Dohare et al., 2021). Before we can combine meta-learning and generate-and-test, we must first develop the generate-and-test approach to auxiliary task discovery so that their combination has the best chance for success. Such an effort is worthy of an entire study on its own, so in this paper we leave combining the two to future work and focus on the generate-and-test approach.

Despite significant interest, it remains unclear what makes a good or bad auxiliary tasks. The meta-learning approaches do not generate human-interpretable tasks. Updating toward multiple previous policies, called the *value improvement path* (Dabney et al., 2020), can improve performance but is limited to historical tasks. The gradient alignment between auxiliary tasks and the main task has been proposed as a measure of auxiliary tasks usefulness (Lin et al., 2019; Du et al., 2018). However, the efficacy of this measure has not been thoroughly studied. Randomly generated auxiliary tasks can help avoid representation collapse (Lyle et al., 2021) and improve performance (Zheng et al., 2021), but can also generate significant interference which degrades performance (Wang et al., 2022).

In this paper we take a step toward understanding what makes useful auxiliary tasks introducing a new generate-and-test method for autonomously generating new auxiliary tasks and a new measure of task usefulness to prune away bad ones. The proposed measure of task usefulness evaluates the auxiliary tasks based on how useful the features induced by them are for the main task. Our experimental results shows that our measure of task usefulness successfully distinguishes between the good and bad auxiliary tasks. Moreover, our proposed generate-and-test method outperforms random tasks, hand-designed tasks, and learning without auxiliary tasks.

## 2 BACKGROUND

In this paper, we consider the interaction of an agent with its environment at discrete time steps $t = 1, 2, \ldots$. The current state is denoted by $S_t \in \mathcal{S}$. The agent's action $A_t \in \mathcal{A}$ is selected according to a policy $\pi : \mathcal{A} \times \mathcal{S} \to [0, 1]$, causing the environment to transition to the next state $S_{t+1}$ emitting a reward of $R_{t+1} \in \mathbb{R}$. The goal of the agent is to find the policy $\pi$ with the highest state-action value function defined as $q_\pi(s, a) \doteq \mathbf{E}_\pi[G_t | S_t = s, A_t = a]$ where $G_t \doteq \sum_{k=0}^\infty \gamma^k R_{t+k+1}$ is called the return with $\gamma \in [0, 1)$ being the discount factor.

To estimate the state-action value function, we use temporal-difference learning (Sutton, 1988). Specifically, we use Q-learning (Watkins & Dayan, 1992) to learn a parametric approximation $\hat{q}(s, a; \mathbf{w})$ by updating a vector of parameters $\mathbf{w} \in \mathbf{R}^d$. The update is as follows,

$$\mathbf{w_{t+1}} \leftarrow \mathbf{w_t} + \alpha \delta_t \nabla_\mathbf{w} \hat{q}(S_t, A_t; \mathbf{w}),$$

where $\delta_t \doteq R_{t+1} + \gamma \max_a \hat{q}(S_{t+1}, a; \mathbf{w_t}) - \hat{q}(S_t, A_t; \mathbf{w_t})$ is the TD error, $\nabla_\mathbf{w} \hat{v}(S_t; \mathbf{w})$ is the gradient of the value function with respect to the parameters $\mathbf{w_t}$, and the scalar $\alpha$ denotes the step-size parameter. For action selection, Q-learning is commonly combined with an epsilon greedy policy.

We use neural networks for function approximation. We integrate a replay buffer, a target network, and the RMSProp optimizer with Q-learning as is commonly done to improve performance (Mnih et al., 2013).

To formulate auxiliary tasks, a common approach is to use general value functions or GVFs (Sutton et al., 2011). GVFs are value functions with a generalized notion of target and termination. More specifically, a GVF can be written as the expectation of the discounted sum of any signal of interest:

$$v_{\pi,\gamma,c}(s) \doteq \mathbf{E}_\pi[\sum_{k=0}^{\infty}(\prod_{j=1}^{k}\gamma(S_{t+j}))c(S_{t+k+1})|S_t = s, A_{t:\infty}\ \pi]$$

where $\pi$ is the policy, $\gamma$ is the continuation function, and $c$ is a signal of interest and is referred to as the cumulant. Similarly, a generalized state-action value function $q_{\pi,\gamma,c}(s,a)$ can be defined where the expectation is conditioned on $A_t = a$ as well as $S_t = s$. A control auxiliary tasks is one where the agent attempts to learn a $\pi$ to maximize the expected discounted sum of the future signal of interest (called a control demon or control GVF in prior work).

To learn these auxiliary tasks, multi-headed neural networks are commonly used where the last hidden layer acts as the representation shared between the main task and the auxiliary tasks (Jaderberg et al., 2016). In this setting, each head corresponds to either the main task or one of the auxiliary tasks and the auxiliary tasks make changes to the representation alongside the main task via backpropagation.

## 3 AUXILIARY TASK DISCOVERY THROUGH GENERATE-AND-TEST

We propose a new method for auxiliary task discovery based on a class of algorithms called generate-and-test. Generate-and-test was originally proposed as an approach to representation learning or feature finding. We can think of backprop with a large neural network as performing a massive parallel search in feature space (Frankle & Carbin, 2018). Backprop greatly depends on the randomness in the weight initialization to find good features. The idea of generate and test is to continually inject randomness in the feature search by continually proposing new features using a *generator*, to measure features usefulness using a *tester*, and to discard useless features. This idea has a long history in supervised learning (Sutton et al., 2014; Mahmood & Sutton, 2013), and can even be combined with backprop (Dohare et al., 2021). The same basic structure can be applied to auxiliary task discovery, which we explain next.

We use generate-and-test for discovering and retaining auxiliary tasks that induce a representation useful for learning the main task. That is, the goal is to find auxiliary tasks that induce a positive auxiliary task effect. It is challenging to recognize which auxiliary tasks induce useful representations. To do so, we first evaluate how good each feature is based on how much it contributes to the approximation of the main task action-value function. Here we define the features to be the output of neural network's last hidden layer after applying the activation function. We then identify which auxiliary task was responsible for shaping which features.

Our proposed generate-and-test method for discovering auxiliary tasks consists of a generator and a tester. The *generator* generates new auxiliary tasks and the *tester* evaluates the auxiliary tasks. The auxiliary tasks that are assessed as useful are retained while the auxiliary tasks that are assessed to be useless are replaced by newly generated auxiliary tasks. The newly generated auxiliary tasks will most likely have low utility. To prevent the replacement of newly generated auxiliary tasks, we calculate the number of steps since their generation and refer to that as their age. An auxiliary task can only be replaced if its age is bigger than some age threshold. Every $T$ time step, some ratio of the auxiliary tasks get replaced. We refer to $T$ as the replacement cycle and denote the replacement ratio by $\rho$. The pseudo-code for the proposed generate-and-test method is shown in Algorithm 1.

Note that the proposed method does not generate-and-test on features but on auxiliary tasks. It, however, does assess the utility of features and derives the utility of the auxiliary tasks from the utility of the features that they induced.

We propose a tester that evaluates the auxiliary tasks based on how useful the features induced by them are for the main task. When following the standard practice of jointly learning the main task and the auxiliary tasks, recognizing which feature was influenced the most by which auxiliary task is challenging. This is because all features are jointly shaped by all the tasks, both auxiliary and main. To address this issue, we use a strategy for learning the representation where all features are used by all tasks in the forward pass; however, each feature is only modified through the gradient backpropagated from one task. See Figure 1. This learning strategy is similar to the Master-User

---

**Algorithm 1** Generate-and-test for auxiliary task discovery

---

1: **Input**: number of auxiliary tasks $n$, age threshold $\mu$, replacement cycle $T$, replacement ratio $\rho$
2: **Initialization:**
3: generate $n$ auxiliary tasks using the *generator*
4: randomly initialize the base learning network
5: set age $a_i$ for each auxiliary task to zero
6: **for** Every time step **do**
7:      do a DQN step to update the base learning network
8:      Increase $a_i$ by one for $i = 1, \ldots, n$
9:      update the utility of each auxiliary task $u^{\text{aux}}(i)$ for $i = 1, ..., n$ using the *tester*
10:      **for** Every $T$ time steps **do**
11:          Find $n\rho$ auxiliary tasks with the lowest utilities such that $a_i > \mu$
12:          replace the $n\rho$ auxiliary tasks with new auxiliary tasks generated by the *generator*
13:          reinitialize the input and output weights of the features induced by the $n\rho$ auxiliary tasks
14:          reset $a_i$ to zero for the $n\rho$ auxiliary tasks

---

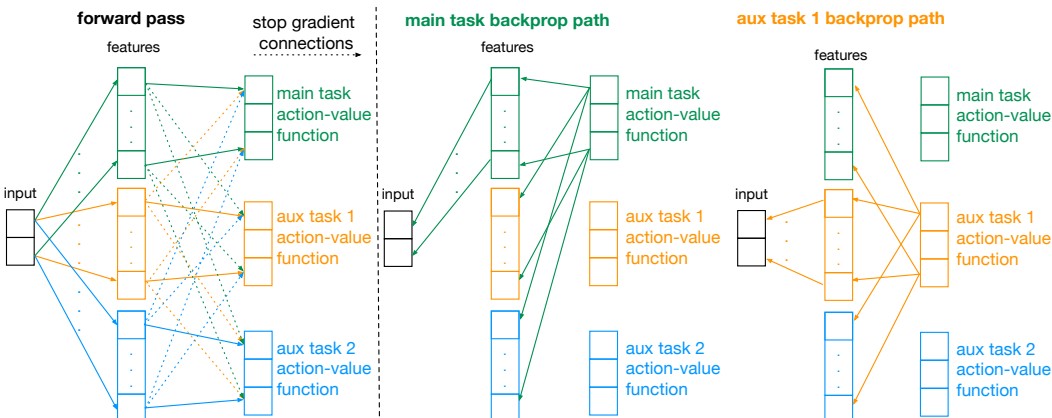

Figure 1: The forward pass, backward pass for the main task, and backward pass for auxiliary task 1 when using the Master-User strategy for learning auxiliary tasks alongside the main task. All features are used by all tasks in the forward pass but only modified through the gradient backpropagated from one task. The dotted arrows show stop-gradient connections. The gradients does not go back any further from these connections. When using the Master-User strategy, it is clear which auxiliary task was responsible for inducing which feature.

algorithm proposed for continual recurrent learning (Javed et al., 2021). Therefore, we refer to this learning strategy as the Master-User strategy. When using the Master-User strategy, it is clear which auxiliary task was responsible for inducing which feature.

As we mentioned above, the proposed tester assesses the utility of an auxiliary task based on how useful the features induced by it are for the main task. To assess each feature, the proposed tester looks at the magnitude of the outgoing weights from the feature to the main task action-value function for all actions. The magnitude of the weights represents how much the feature contributes to the approximation of the main task action-value function. The greater the magnitude is, the more important the feature is. The tester also considers the magnitude of each feature: the greater the magnitude of the feature is, the more it contributes to the approximation of the main task action-value function. Therefore, the instantaneous utility of a feature $f_k^i$ is defined as:

$$u(f_k^i) = \sum_a |w_{ka}^{\text{main}} \times f_k^i| \tag{1}$$

where $f_k^i$ is the $k$th feature shaped by auxiliary task $i$ and $u(f_k^i)$ is the instantaneous utility of feature $f_k^i$. Instead of only looking at the instantaneous utility $u(f_k^i)$, the proposed tester considers a *trace* of the past utilities using an exponential moving average of $u(f_k^i)$'s:

$$\bar{u}(f_k^i) \leftarrow (1 - \tau)\bar{u}(f_k^i) + \tau u(f_k^i) \qquad (2)$$

where $\bar{u}(f_k^i)$ is a trace of $u(f_k^i)$ with the trace parameter denoted by $\tau$. This assessment method is similar to what has been used in generate-and-test on features (Mahmood & Sutton, 2013).

After assessing the utility of the features, the utility of each auxiliary task is set to the sum of the utility of the features shaped by it: $u^{\text{aux}}(i) = \sum_k \bar{u}(f_k^i)$.

We combined the proposed tester with a simple generator that randomly generates auxiliary tasks. The auxiliary task are formulated as *subgoal-reaching* GVFs where the continuation function returns 0 at the subgoals and 1 elsewhere (similar to $\gamma$ in an epsiodic MDP). The cumulant is $-1$ everywhere and the policy is greedy. In plain english, b The subgoals are randomly selected from the observation space, meaning the agent is learning many policy to reach different parts of the observation space in addition to solving the main task.

## 4 EXPERIMENTAL RESULTS

In this section, we provide empirical results supporting the efficacy of the proposed generate-and-test method for auxiliary task discovery. We include results on two gridworld environments: four-rooms and maze. We also include results on the pinball environment (Konidaris & Barto, 2009), which is widely used in skill chaining, option discovery, and recently model-based planning (Lo et al., 2022). We choose these environments so that we could easily visualize the discovered auxiliary tasks and easily design good and bad auxiliary task as baselines. All environments are episodic.

In the gridworld environments, the goal is to learn the shortest path from start state to goal. The start and goal states are denoted by S and G respectively in Figure 2 and 3. At each cell, four actions are available: up, down, left, and right, which moves the agent one cell in the respective direction. The observation space is described with a one-hot representation with the index corresponding to the agent's position being 1. The reward is $-1$ on each time step. There is an episode cutoff of 500 steps.

In the pinball environment, a small ball should be navigated to the goal in a maze-like environment with simplified ball physics. In Figure 3, the pinball environment is shown with the ball and goal shown by a grey and yellow circle respectively. Collision with the obstacles causes the ball to bounce. The observation space is continuous and is described by $x, y, \dot{x}, \dot{y}$. The start location and goal location are at $(0.8, 0.5)$ and $(0.1, 0.1)$ respectively. The action space includes 5 actions of increasing or decreasing $\dot{x}$ or $\dot{y}$ and no change to $\dot{x}$ and $\dot{y}$. The reward is $-5$ at each time step. There is no episode cutoff.

Note that in the original pinball environment, the agent receives a special reward of $10,000$ upon arrival at the goal. Instead we gave a reward of -5 (like every other step) so that the scale of the action-value function for the main task and the auxiliary tasks would not be too different. When learning multiple tasks in parallel, the contribution of each task is determined by the scale of the corresponding value function (Hessel et al., 2019). Therefore, when the scale of value functions are very different, we would need to scale the reward of the main task and the cumulants of the auxiliary tasks appropriately. This issue requires an additional hyper parameter that would give our method an advantage if tuned. For this paper, we decided to focus on the case where the scale of the value function for the main task and the auxiliary tasks are similar.

We used DQN with RMSProp optimizer as the base learning system. We used a neural network with one hidden layer and $tanh$ activation function. (We used $tanh$ activation function so that the induced features would be all in the same range of $(-1, 1)$; however, our proposed tester should work well when other activation functions are used too. This can be investigated in future work.) For the girdworld environments, the one-hot observation vector was fed to the neural network. The hidden layer size for four-rooms and maze were 50 and 500 respectively. The replay buffer size for

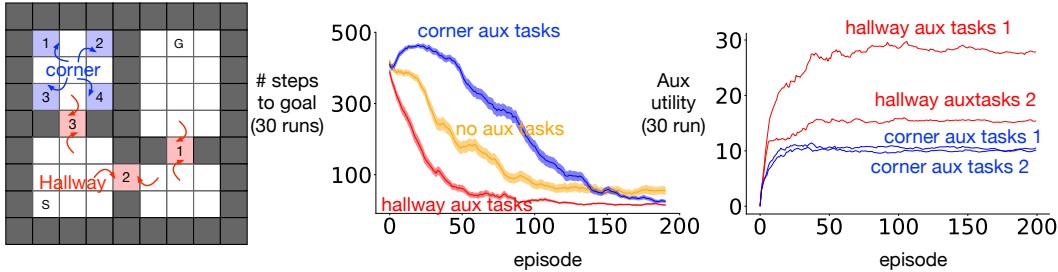

Figure 2: Left: The four-rooms environment with the good and bad hand-designed auxiliary tasks shown in red and blue respectively. Middle: Hallway and corner auxiliary tasks improved and hurt the performance respectively. Right: The proposed tester evaluated the hand-designed auxiliary tasks well, giving higher utility to the hallway auxiliary tasks.

four-rooms and maze were 500 and 1000 respectively. For both four-rooms and maze, we used a batch-size of 16 and target network update frequency of 100.

For the pinball environment, the 4-dimensional observation was normalized and fed to the neural network. The hidden layer size was 128. We used a replay buffer of size $10,000$, a batch-size of 16, and target network update frequency of 200.

## 4.1 THE PROPOSED TESTER REASONABLY EVALUATES THE AUXILIARY TASKS

To see how well the proposed tester evaluates the auxiliary tasks, we designed good and bad auxiliary tasks in the four-rooms environment. The hand-designed auxiliary task were formulated as subgoal-reaching GVFs with the good and bad hand-designed auxiliary tasks having hallway and corner subgoals respectively. See Figure 2.

Note that when learning the auxiliary tasks alongside the main task using the Master-User strategy, the gradient backpropagated from the main task only modifies $\frac{1}{\text{auxiliary tasks}+1}$ percent of the features. For example, in the case of learning the hallway auxiliary tasks, there are 3 auxiliary tasks. Therefore, the gradient backpropagated from the main task only modifies 25% of the features.

The hallway and corner auxiliary tasks improved and hurt learning in terms of learning speed respectively as expected (Figure 2, middle graph). The proposed tester evaluated the hallway and corner auxiliary tasks well, assigning higher utility to the hallway auxiliary tasks and clearly indicating the corner tasks are bad.

## 4.2 THE GENERATE-AND-TEST METHOD IMPROVES OVER THE BASELINE OF NO AUXILIARY TASKS

Next, we studied the performance of the base learning system when combined with the proposed generate-and-test method. The generate-and-test method uses the combination of the *random generator* and our proposed tester. The random generator produces subgoal-reaching auxiliary tasks with the subgoals randomly picked from the observation space. More specifically, in the gridworld environments, the subgoals are cells in the grid. In the pinball environment, the subgoals are determined by $(x, y)$ and once the ball is within radius $0.035$ of a subgoal, it is assumed that the agent has reached the subgoal.

We included four baselines for comparison which included the base learning system with 1) no auxiliary tasks 2) hand-designed good auxiliary tasks 3) hand-designed bad auxiliary tasks 4) fixed random auxiliary tasks. All the auxiliary tasks were in form of subgoal-reaching tasks. The subgoals corresponding to the hand-designed good and bad auxiliary tasks for all three environments are shown in red and blue respectively in Figure 3. For the fixed random auxiliary tasks, the subgoals where randomly picked from the observation space and kept fixed throughout learning.

We systematically swept the step-size parameter and report the performance of the best to ensure a fair comparison. To do so, we ran the baseline with no auxiliary tasks with different values of the step-size for 10 runs. We used the step-size that resulted in the lowest area under the curve and reran the baseline with the best step-size for 30 runs to get the final results. We repeated this process for the baselines with hand-designed auxiliary tasks. For the generate-and-test method, we used the same step-size as the baseline with good hand-designed auxiliary tasks. For four-rooms, maze, and pinball the sweep over the step-sizes included $\{0.000625, 0.0025, 0.01, 0.04\}$, $\{0.00025, 0.001, 0.004\}$, and $\{0.0025, 0.005, 0.01\}$.

The generate-and-test method has hyper-parameters of its own: 1) number of auxiliary tasks 2) age threshold 3) replacement cycle 4) replacement ratio. For the gridworld environments, we used 8 auxiliary tasks, age threshold of 0, replacement rate of 1000 steps, and replacement ratio of 0.25. For the pinball environment, we used 5 auxiliary tasks, age threshold of 5000, replacement rate of 5000 steps, and replacement ratio of 0.2.

The proposed generate-and-test method outperformed the baseline with no auxiliary tasks in all three environments (Figure 3). The generate-and-test method also outperformed the baseline with fixed random auxiliary tasks. This suggests that the subgoals discovered by the generate-and-test are actually better than random subgoals.

Interestingly, the fixed random auxiliary tasks resulted in performance gain over the baseline with no auxiliary tasks in all three environments (Figure 3). This is in line with the findings from the literature suggesting that random GVFs can form good auxiliary tasks for reinforcement learning (Zheng et al., 2021).

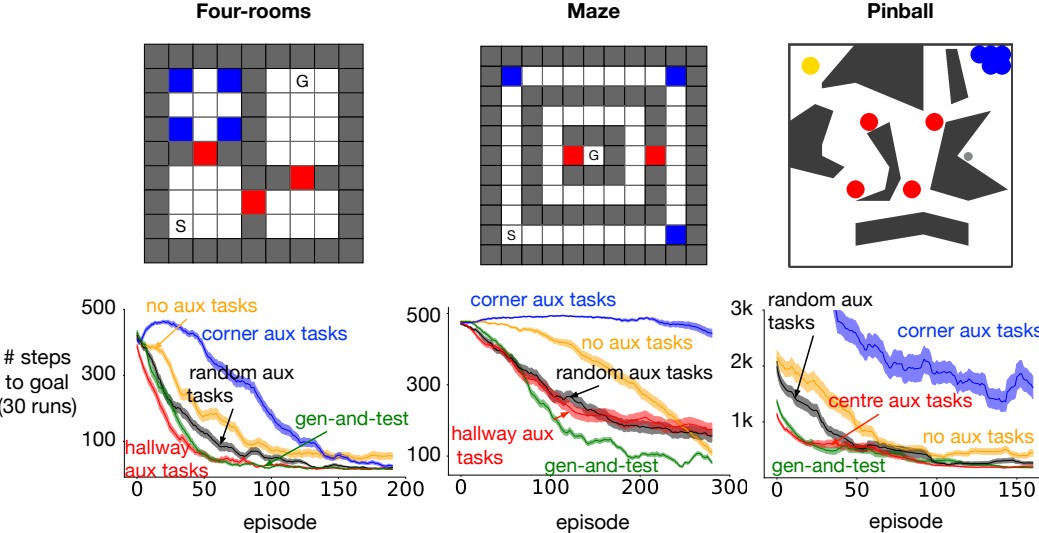

Figure 3: The learning curves for the proposed generate-and-test method (green), the baseline with no auxiliary tasks (orange), the baseline with fixed random auxiliary tasks (black), and the baseline with good and bad hand-designed auxiliary tasks (red and blue). The results are averaged over 30 runs. The proposed generate-and-test method improved over the baseline with no auxiliary tasks. The hand-designed good and bad auxiliary tasks respectively improved and hurt the performance. Generate-and-test also outperformed the baseline with fixed random auxiliary tasks. Fixed random auxiliary tasks also resulted in performance gain over the baseline.

### 4.3 THE AUXILIARY TASKS DISCOVERED BY THE GENERATE-AND-TEST METHOD ARE REASONABLY GOOD

We conducted additional experiments to analyze how good the auxiliary tasks discovered by the generate-and-test method are. In the previous subsection, we compared the performance of the base learning system combined with generate-and-test and combined with fixed random auxiliary

tasks. Generate-and-test outperformed fixed random auxiliary tasks(Figure 3). This suggests that the choice of the auxiliary tasks was important and generate-and-test discovered and retained useful auxiliary tasks.

The auxiliary tasks discovered and retained by generate-and-test are shown in Figure 4. To plot the discovered auxiliary tasks, we ran the generate-and-test method for 30 runs and stored the auxiliary tasks that were retained. The green squares correspond to the discovered auxiliary tasks in the gridworld environments. Darker green indicates that the state was chosen as a subgoal in many runs. For the pinball environment, the discovered auxiliary tasks are shown by green circles. In the gridworld environments, the subgoals corresponding to the discovered auxiliary tasks were close to the goal states. In the pinball environment, the discovered auxiliary tasks were more concentrated in the central areas—reasonable way-points on the path to the goal.

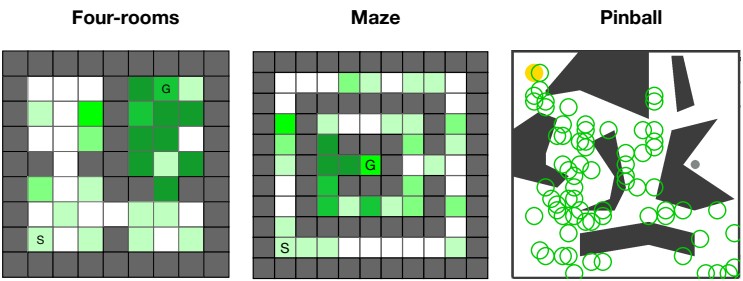

Figure 4: Example discovered auxiliary tasks in the three environments. Generate-and-test discovered reasonably good auxiliary tasks: In the gridworld environments, the subgoals corresponding to the discovered auxiliary tasks were close to the goal states. In the pinball environment, the discovered auxiliary tasks were more concentrated in the central areas.

To confirm that the auxiliary tasks discovered by generate-and-test were useful, we stored the auxiliary tasks discovered and retained, in a pool. We then randomly selected a number of auxiliary tasks from the pool and ran the base learning system, learning the main value function from scratch. We kept the auxiliary tasks fixed throughout learning. We repeated this for 30 runs. The discovered auxiliary tasks were useful and substantially improved over the baseline of no auxiliary tasks (Figure 5).

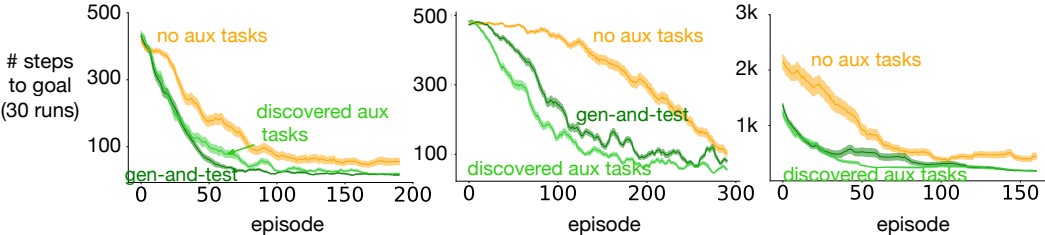

Figure 5: The learning curves corresponding to the discovered auxiliary tasks. The discovered auxiliary tasks were kept fixed throughout learning. The discovered auxiliary tasks improved over the baseline of no auxiliary tasks.

### 4.4 THE REPRESENTATION INDUCED BY THE GENERATE-AND-TEST METHOD HAS LOWER REDUNDANCY COMPARED TO THE BASELINE.

Next, we studied the representation that emerged under the generate-and-test method. There are a multitude of metrics for evaluating the emergent representations (Wang et al., 2022; Javed & White, 2019). We used the stable rank of the weight matrix between the input layer and the hidden layer (Arora et al., 2018). The stable rank of a matrix A is defined as $\frac{\sum_i \sigma_i^2}{max_i \sigma_i^2}$ where $\sigma_i$ are the the singular values of matrix $A$. The stable rank provides an approximation of the rank of the matrix but it is

unaffected by the smaller singular values. The stable rank of the weight matrix between the input layer and the hidden layer characterizes the amount of generalization/redundancy of the network. The larger the stable rank is, the lower the redundancy in the representation is.

The stable rank of the representation learned by generate-and-test is larger than the stable rank of the representation learned by the baseline with no auxiliary tasks (Figure 6). This suggests that the auxiliary tasks discovered by generate-and-test resulted in a representation with lower redundancy. The lower stable rank of the learned representation together with the better performance is encouraging and suggests that more aggressive pruning by the tester could get a lower stable rank while maintaining good performance.

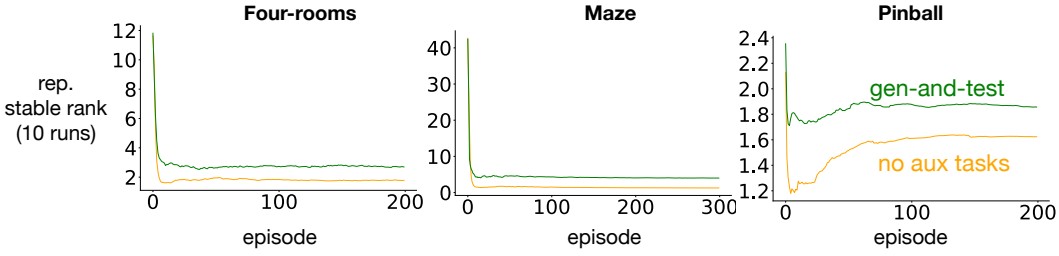

Figure 6: The stable rank of the weight matrix from the input layer to the hidden layer over episodes. The results are averaged over 10 runs. The combination of the base learning system with the generate-and-test method resulted in a representation with a higher stable rank compared to the case of only having the base learning system. This suggests that the auxiliary tasks discovered by generate-and-test resulted in a representation with lower redundancy.

## 5 CONCLUSIONS AND FUTURE WORK

In this paper, we proposed a new method for auxiliary task discovery. The proposed method uses a generate-and-test approach. We also introduced a new measure of auxiliary task usefulness. Through careful experimentation, we showed that: 1) The proposed tester reasonably evaluates the auxiliary tasks. 2) The generate-and-test method improves over the baseline with no auxiliary tasks. 3) The auxiliary tasks discovered by the generate-and-test method are reasonably good. 4) The representation induced by the auxiliary tasks discovered by generate-and-test has lower redundancy compared to the baseline with no auxiliary tasks.

An interesting future work would be to investigate different generators and testers for auxiliary tasks. For example, to improve the generator, one idea is to to sample subgoals from the replay buffer instead of generating random subgoals. This would be similar to the idea of hindsight experience replay (Andrychowicz et al., 2017). Another idea to test for the generator is to generate feature-attainment auxiliary tasks instead of subgoal-reaching auxiliary tasks. For feature-attainment auxiliary tasks, the goal is to maximize a feature of the state representation (Sutton et al., 2022). In this paper, we took a first step toward designing a functional generate-and-test method for auxiliary task discovery. However, there is a big space of ideas to try for designing the generator and the tester.

Another future direction is to combine the meta-learning approaches with the proposed generate-and-test method. The meta-learning approaches can be used to refine the auxiliary tasks discovered by generate-and-test. It would be interesting to test the combination of meta-learning with generate-and-test in the three tested environments and see if their combination will result in discovering good auxiliary tasks, outperforming both approaches in isolation.

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
