# OpenReview forum: "Auxiliary task discovery through generate and test"
_ICLR.cc/2023/Conference — Submitted to ICLR 2023_

### Official Review · Reviewer_CKEP · 2022-10-19

**Confidence:** 5
**Correctness:** 3
**Technical Novelty And Significance:** 3
**Empirical Novelty And Significance:** 2
**Recommendation:** 5

**Clarity, Quality, Novelty And Reproducibility:**

### **Quality**
The quality of this paper is above average. However, the evaluation is not thorough enough. The environments are too simple to investigate the correctness of the proposed method in more complicated situations.

### **Clarity**
The presentation of this paper is good. Section 3 may need to be improved with re-organization.

### **Originality**
As far as I know, using the weight of NN as an indicator to generate tasks for RL agents is an original idea.

### **Reproducibility**
The code is provided and the idea is easy to implement. I think the proposed method is reproducible.


**Strength And Weaknesses:**

### **Strength**

* This paper tackles a very interesting idea. Using auxiliary tasks to augment NN has shown improvement in the supervised learning area. This paper uses this idea to reinforcement learning.

* The writing of this paper is generally good. The idea is neat and easy to understand. Figure 1 helps me understand the framework.

### **Weaknesses**

* **Clarification.**
    * Title is not concrete. According to Section 2, the paper uses the setting of MDP. I think the title should be more concrete by adding reinforcement learning or q-learning since the authors mention that they use q-learning as the base framework.
    * How to measure the influence of auxiliary tasks is the key to the entire paper. A high-level idea of this metric (in my understanding, using the magnitude of features for the main task as an indicator) should be mentioned earlier, for example, in the abstract or introduction.
    * Section 3 can be improved by using subsections to separate contents, for example, generate and test framework, tasks generator, tasks evaluation, etc. The current version put all things together, which makes it hard to find important information.

* **Generalization to other tasks.** At the end of Section 3, the authors discuss the design of the task generator. It seems that the generator just randomly selects sub-goals in the observation space (similar to some curriculum learning algorithms). How to extend this generator to other tasks? Do we still human to design the candidate task?

* **The correctness of the proposed metric.** Following last point, I also suspect that the proposed metric for evaluating the usefulness of tasks only works in limited cases. In the experiments of this paper, the features of different goals may be very different, thus using the magnitude is a good approximation. However, in more complicated situations, such as image input tasks or continuous control tasks, most of the features (representing dynamics or semantic objects) are shared across all tasks, including both good and bad ones. In these environments, I suspect that the magnitude does not make sense. More experiments on complicated environments should be done to investigate this point.


* **Questions to experiments.**
	* In the middle part of Figure 2, why does the curve of corner aux tasks finally reach the same level as the curve of hallway aux tasks? I expect that the bad aux tasks should harm the learning. I guess one explanation is that adding more goals always enables the agent to solve the sparse reward problem as stated in Hindsight Experience Replay (HER)[1].
	* In the middle part of Figure 3, the curve of generate-and-test is much better than the curve of hallway aux tasks. It is a little surprising since hallway aux tasks are reasonable middle points to the goal.
	* In Figure 4, it seems that useful tasks are all close to the goal. In long-horizon problems, these tasks still have the sparse reward problem. The intuition behind these tasks is not clear.


---

[1] Andrychowicz, M., Wolski, F., Ray, A., Schneider, J., Fong, R., Welinder, P., McGrew, B., Tobin, J., Pieter Abbeel, O. and Zaremba, W., 2017. Hindsight experience replay. Advances in neural information processing systems, 30.


**Summary Of The Paper:**

This paper focuses on designing auxiliary tasks that could tend to improve data efficiency by forcing the agent to learn auxiliary prediction and control objectives in addition to the main task. Previous work usually designs auxiliary tasks with human knowledge, which are computationally expensive and challenging to tune in practice.
In this paper, the authors propose continually generating new auxiliary tasks and preserving only those with high utility. They introduce a new measure of auxiliary tasks’ usefulness based on how useful the features induced by them are.
Experiments on the Gridworld and the Pinball environments show that their method outperforms random tasks, hand-designed tasks, and learning without auxiliary tasks.


**Summary Of The Review:**

My biggest concern is the proposed method does not fit more complicated tasks. Using the magnitude of features seems to only work in limited situations. I suggest rejecting this paper unless more evidence is provided.

---

> ### Author Response · Authors · 2022-11-08
> **Response to Reviewer CKEP**
>
> Thank you for the helpful comments!
>
> **Regarding generalization to other tasks:**
>
> For improving/scaling the generator, there is a multitude of exciting ideas to try. One idea is to sample subgoals from the replay buffer. This would be similar to the idea of hindsight experience replay (HER) (Andrychowicz et al., 2017). You can find the result for a preliminary experiment that we did to test this idea in 'supplementary material/Author response figures/3.png'. The generator samples subgoals from the replay buffer with the samples with higher TD-error having a higher chance of getting selected. The generate-and-test method with HER generator outperformed the baseline with no auxiliary tasks. This is promising because the HER generator is less prone to scalability issues compared to the random generator. Of course, further experimentation is required to fully understand how robust the HER generator is. We would like to emphasize that we are at the beginning of exploring a whole new approach to auxiliary task discovery through generate and test. While there is a lot more to do to improve our generator and tester, especially in terms of scalability, we believe that we took a small but meaningful step toward automating the process of auxiliary task discovery. We showed that even with a random generator, reasonably good auxiliary tasks can be discovered in small domains by the generate-and-test approach. With smarter generators, we could hope to achieve even better results.
>
> **Regarding the correctness of the proposed metric:**
>
> Results from the literature suggest that different auxiliary tasks induce different features in complicated domains. If the features induced by all tasks, auxiliary or main, were the same, there would be no difference between the auxiliary tasks in terms of usefulness. The results in the literature, however, suggest otherwise: some auxiliary tasks result in substantial performance gain over the baselines whereas others harm the performance (shelhamer et al., 2016).
>
> **Answers to the questions about the experiments:**
>
> - Corner auxiliary tasks are harming the performance by harming the data-efficiency/speed of learning. Regarding why the baseline with corner auxiliary tasks reach a good level of performance in the end, we emphasize that the features induced by the main task are not affected by the corner auxiliary tasks and for the four-rooms environment, those features were sufficient for learning.
> - As shown in the learning curve, the doorway auxiliary tasks outperform the baseline with no auxiliary task, so your intuition about doorway auxiliary tasks being reasonable middle points to the goal is correct. However, the auxiliary tasks found by our generate-and-test method are even better. We emphasize that, as shown in Figure 4, the inner doorway is discovered and recognized as useful by our method.
> - We conjecture that in long-horizon problems, the auxiliary tasks discovered by our method will not be necessarily close to the goal. For example even in the pinball environment, which has a longer horizon compared to the gridworld domains, the discovered subgoals are not all distributed around the goal state.

---

### Official Review · Reviewer_kAkb · 2022-10-24

**Confidence:** 3
**Correctness:** 3
**Technical Novelty And Significance:** 2
**Empirical Novelty And Significance:** 2
**Recommendation:** 5

**Clarity, Quality, Novelty And Reproducibility:**

- In general, the presentation is somewhat clear, but I see some issues (please see my comment above).
- The overall quality could be improved in terms of supporting the main claim, presenting the alignment of the main task performance improvements and the measured auxiliary utilities (please see my comment above)
- In terms of the originality, I see some limited novelty in the measurement of the usefulness of each feature by using the scale of its contribution to the main value function.

**Strength And Weaknesses:**

Strengths

- In the used environments and tasks, the proposed method seems to select helpful subgoals in the state space for subgoal-reaching auxiliary tasks.
- The description of the method is concise and easy to understand. Especially, Alg.1 and Fig.1 are quite helpful for an overview of the proposed approach.

Weaknesses

- I'm mainly concerned about the soundness of the proposed method. If I'm not mistaken, except for the features dedicated to the main task, all the features are connected with the value function for the main task via random weights that are not updated with backpropagation during the training. In my opinion, it means that some auxiliary features might contribute large-scale values to the main value function but with small relevance to the main task, so that some of the main features are trained to "undo" (or compensate for) the irrelevant contribution.
- I think the writing could be improved in some ways. For instance, the measurement of the usefulness of features based on the scales of the contributed values could be summarized and introduced as an overview in the abstract or at least Sec.1 (Introduction). There is a typo regarding the gradient notation in Sec.2. A typo "auxtasks" is in Fig.2.
- I see some issues in the presentation. In Fig.2, the Aux utility plots for "hallway aux tasks 3" and "corner aux tasks 3-4" are missing, which makes the figure less convincing. Also, presenting and comparing the heat maps of the performance improvements and the measured auxiliary utilities in the state space by testing many more subgoals could be quite informative.
- Empirically, I don't fully agree that the proposed algorithm "significantly outperforms" hand-designed tasks. One issue I see in this regard is that since subgoal-reaching auxiliary tasks are simple and easy to create unless the state space is huge, in the given environments and tasks, it may often be enough to form auxiliary tasks by hand.

**Summary Of The Paper:**

The authors propose a generate-and-test approach to discovering auxiliary tasks in RL. They form a multi-headed network where each head outputs a (general) value function for each task. At the last layer, the gradient update is only applied to the weights between each head and each dedicated set of features, and the rest of all the weights are used only in forward passes. The authors compute the magnitude of the value that each feature contributes to the value function for the main task and use it as a measure of how helpful that feature is to the main task. They evaluate their approach in the Four rooms, Maze, and Pinball environments.

**Summary Of The Review:**

While I appreciate their concise description of the method and the meaningful empirical results, I have multiple concerns. My major concern is about the soundness of the method: each utility is measured by involving fixed random weights and their contributions to the main value function could be undone by some of the main features. I have some suggestions regarding the writing and presentation (especially of the experiments) as well.

---

> ### Author Response · Authors · 2022-11-08
> **Addressing Reviewer kAkb main concerns**
>
> Thank you for the helpful comments!
>
> **Regarding the concern about the soundness of the proposed method:**
>
> We emphasize that the main task does make changes to the **weights** between itself and a feature whose master is an auxiliary task; however, it does not change the **feature** itself. More specifically, the weights from each feature to each task are learned by gradient backpropagated from that task. However, the weights from the input layer to a feature is only learned through the gradient backpropagated from that feature’s master task. We have provided the gradient path for the main task and auxiliary task 1 as examples in a figure in 'supplementary material/Author response figures/1.png'. You can see in the figure on the left that the gradient path for the main task, shown in green, includes the connections between the main task and all features. However, it only includes the connections between the input and the green features. **Therefore, the backward pass starting from the main task action-value function does not change the input weights to the orange features.**
>
> This should address your main concern regarding the correctness of the proposed method. We will change the text of the paper to make the explanation of the Master-User architecture more clear.
>
> **Regarding Figure 2 missing the curves corresponding to two of the corner auxiliary tasks and one of the hallway auxiliary tasks:**
>
> In the results reported in Figure 2, 4 hand-designed (2 corner + 2 hallway) auxiliary tasks are learned in parallel alongside the main task. We reran the experiment such that all 7 hand-designed (4 corner + 3 hallway) auxiliary tasks are learned in parallel alongside the main task. You can find the new results in 'supplementary material/Author response figures/2.jpeg'. The results are as expected: the tester assigned higher scores to the hallway auxiliary tasks. Note that in this case each auxiliary task only modifies 50/(7+1) ~ 6 features whereas in the results reported in Figure 2, each auxiliary task modified 50/(4+1) ~ 10 features. Therefore, the results of the two cases are not comparable.
>
> **Regarding the last comment in the Weaknesses section:**
>
> As you said, in small environments, auxiliary tasks could be easily hand-designed; however, in larger environments relying on human experts for designing them is not feasible. That is why automating the discovery of auxiliary tasks is important. In this paper, we took a small but meaningful step toward automating the discovery of auxiliary tasks and proposed a reasonable measure of auxiliary task usefulness which is applicable to any domain. One of the main reasons why we experimented with small domains was exactly that it is easy for the human designer to find good and bad auxiliary tasks in small domains. Therefore, we could have good and bad baselines to compare our method to. In all of our experiments, the auxiliary tasks discovered by our method outperformed the baseline with hand-designed bad auxiliary tasks and matched or outperformed the baseline with hand-designed good auxiliary tasks.

---

> > ### Author Response · Authors · 2022-11-10
> > **Kindly requesting Reviewer kAkb to acknowledge our clarification regarding the soundness of the proposed method**
> >
> > Given that the discussion period is short and we made clarifications to the description of the proposed method which had resulted in misunderstandings, we would appreciate it if you could acknowledge that you read our response regarding your main concern about the soundness of the proposed method.

---

> > ### Comment · Reviewer_kAkb · 2022-11-16
> > **Response to Authors**
> >
> > Thanks for the response from the authors.
> >
> > - Regarding the soundness of the proposed method, I think the clarified method basically makes sense. Considering this, I'm raising my score to 5. However, I recommend the authors to update the manuscript, especially Fig.1, to be clearer.
> > - About the hand-designed tasks vs the tasks learned by the proposed method, I still believe there should be empirical results in non-trivial and more complex environments. To define the generator for the proposed method, the space of the tasks and how to sample tasks from it should be hand-designed. It means that if the method only works on small or simple tasks, the benefit that this method brings may not be very clear. Thus, in order to show the effectiveness of the proposed method, the authors should present experimental results that suggest that the method can find helpful auxiliary tasks efficiently even with big task search spaces and complex environments.

---

> > > ### Author Response · Authors · 2022-11-16
> > > **Response to Reviewer kAkb**
> > >
> > > Thanks for the response. We appreciate you getting back to us before the end of the discussion period!
> > >
> > > We updated Figure 1 and the text of the paper to address the ambiguity in the description of the Master-User strategy and uploaded the revised version.
> > >
> > > For improving/scaling the generator, there is a multitude of exciting ideas to try. One idea is to sample subgoals from the replay buffer. This would be similar to the idea of hindsight experience replay (HER) (Andrychowicz et al., 2017). You can find the result for a preliminary experiment that we did to test this idea in 'supplementary material/Author response figures/3.png'. The generator samples subgoals from the replay buffer with the samples with higher TD-error having a higher chance of getting selected. The generate-and-test method with HER generator outperformed the baseline with no auxiliary tasks. This is promising because the HER generator is less prone to scalability issues compared to the random generator. Of course, further experimentation is required to fully understand how robust the HER generator is. We would like to emphasize that we are at the beginning of exploring a whole new approach to auxiliary task discovery through generate and test. While there is a lot more to do to improve our generator and tester, especially in terms of scalability, we believe that we took a small but meaningful step toward automating the process of auxiliary task discovery. We showed that even with a random generator, reasonably good auxiliary tasks can be discovered by the generate-and-test approach in small domains. With smarter general-purpose generators, like the HER generator, we could hope to achieve good results in larger environments as well.

---

### Official Review · Reviewer_Si75 · 2022-10-25

**Confidence:** 5
**Correctness:** 4
**Technical Novelty And Significance:** 3
**Empirical Novelty And Significance:** 3
**Recommendation:** 8

**Clarity, Quality, Novelty And Reproducibility:**

The paper presents a clear problematic and a clear idea to tackle the approach. The paper is well situated with respect to previous work, although more connections to options would be important as auxiliary tasks (GVFs) and options are very closely related. It is mentioned a few times that generate and test could be combined with meta-learning, but this never happens in the paper so it is not clear why such references are made throughout the paper.

The paper claims that meta-learning is not sample efficient, which can be true, but is not always the case. It really depends on what exactly we are meta-learning. For example meta-learning the whole update rule can be costly, but only learning a part of it is much more efficient. There are papers on both sides of the spectrum and it would be a more fair assessment of meta RL to make these references.

The question of evaluating how good an auxiliary task is is a very difficult thing to do, as we can hardly find a good measure. The size of weights proxy is an interesting avenue, but how robust is it? The evaluated environments are smaller in size (especially FourRooms, isnt the standard FourRooms almost twice the size?), so it is not clear how this can scale up. More importantly, how can defining random task scale up beyond the one-hot representations? What about images are inputs? This could be done on these environments. A more detailed connection with Hindsight Experience Replay would also be important.

**Strength And Weaknesses:**

# Strength
- Clear presentation and motivation
- Solid and trust-worthy empirical investigation
- A simple method that seems to do well on the environments tested.

# Weaknesses
- Concerns in terms of scalability of the method beyond one-hot representations
- Additional baselines would be a very good addition to the paper
- A lot of additional hyperparameters

**Summary Of The Paper:**

The paper proposes a way to generate auxiliary predictions through the generate and test mechanism. The predictions are randomly generated and are then evaluated for their utility by a proxy measure: by looking at the magnitude of the weights between the features and the the main task learner. The authors claim that this approach is somewhere in-between end-to-end learning, where sample efficiency is a concern and hand-specified tasks. The authors report improved sample efficiency on three environments that are smaller in scale. Ablation studies highlight important properties of the algorithm.

**Summary Of The Review:**

The paper proposes an interesting way to generate and test random predictions and evaluate the sample efficiency. The experiments are informative, but still leave a bit of doubt in terms of scalability.

---

> ### Author Response · Authors · 2022-11-08
> **Response to Reviewer Si75**
>
> Thank you for the helpful comments!
>
> **Regarding the scalability of the proposed method:**
>
> As you mentioned, the problem of evaluating and discovering auxiliary tasks is very challenging.
> We would like to emphasize that we are at the beginning of exploring a whole new approach to auxiliary task discovery through generate and test. While there is a lot more to do to improve our generator and tester, especially in terms of scalability, we believe that we took a small but meaningful step toward automating the process of auxiliary task discovery. We showed that even with a random generator, reasonably good auxiliary tasks can be discovered by the generate-and-test approach. With smarter generators, we could hope to achieve even better results.
>
> Regarding the scalability of the tester, we agree that further experimentation is required. However, given that the principle behind the tester, evaluating the auxiliary tasks based on how useful the features induced by them are for the main task, is general and applicable to any task, we think the proposed tester is promising and worth pursuing. We should also note that the pinball environment has a continuous state space and the state representation used for it is not one-hot.
>
> Regarding the scalability of the generator, we agree that random subgoals will not work in larger environments. For improving the scalability of the generator, there is a multitude of exciting ideas to try. One idea is to sample subgoals from the replay buffer. This would be similar to the idea of hindsight experience replay (Andrychowicz et al., 2017). You can find the result for a preliminary experiment that we did to test this idea in 'supplementary material/Author response figures/3.png'. The generator samples subgoals from the replay buffer with the samples with higher TD-error having a higher chance of getting selected. The generate-and-test method with HER generator outperformed the baseline with no auxiliary tasks. This is promising because the HER generator is less prone to scalability issues compared to the random generator. Of course, further experimentation is required to fully understand how robust the HER generator is.
>
> **Regarding meta-gradient sample-efficiency:**
>
> We were specifically referring to the case where the cumulant and continuation function of the auxiliary tasks are learned by non-mypoic meta-gradients (Veeriah et al., 2019). The computation of non-mypoic meta-gradients requires multiple unrolling steps in order to compute the effect of changing the auxiliary tasks over multiple steps. This is computationally expensive. We will change the writing of the paper to make our statement about meta-gradient sample-efficiency more clear.

---

> > ### Comment · Reviewer_Si75 · 2022-11-17
> > **Response to the author rebuttal**
> >
> > I would like to thank the authors for taking the time to write a thoughtful rebuttal.
> >
> > If I understand correctly, the crucial contribution of the paper is the tester. After looking into the additional results where the HER generator is used, it does seem like the tester is compatible with different generators. This result is good evidence and definitely supports the paper. However, there is still an important baseline that has to be considered. If I understand correctly, the "random aux tasks" generates random predictions only once at the start of training, while the proposed method gets to generate-and-test new random predictions as the training progresses. I think an important baseline would be to generate random predictions every N steps (where N is the same amount of steps used by the proposed algorithm) and randomly replace some of the existing predictions. This would be a clear demonstration that the tester is actually necessary to obtain improved performance. I understand that this might not be possible to complete at the moment, but it is a distinction that I find quite important to investigate.
> >
> > Regarding meta gradients, I understand what the authors mean. I would like to encourage them to give a broader statement about meta learning in the paper, as meta learning is not limited to the non-myopic updates of the referenced paper.

---

> > > ### Author Response · Authors · 2022-11-17
> > > **Response to Reviewer Si75**
> > >
> > > Thanks a lot for the response and the great suggestion.
> > >
> > > We are running the experiments for the baseline that you suggested and will upload the results if they are finished by the end of the discussion phase.
> > >
> > > We will included a detailed discussion of the data efficiency and computational complexity of meta-gradient with non-myopic learning objectives in the camera ready version of the paper.

---

> > > ### Author Response · Authors · 2022-11-18
> > > **Providing the results for the baseline suggested by Reviewer Si75**
> > >
> > > We ran the baseline that you suggested, generate and test with a random tester, in all three environments. For this baseline, every T step, a percentage of the auxiliary tasks get replaced randomly. More specifically, every T step, $n \times \rho$ auxiliary tasks get replaced where n is the number of auxiliary tasks and $\rho$ is the replacement ratio. We used the same values for T, n, and $\rho$ as the generate-and-test method with the proposed weight-magnitude tester.
> > >
> > > You can find the results in 'supplementary material/Author response figures/4.pdf. The results are averaged over 30 runs. The generate-and-test method with our proposed tester outperformed the generate-and-test method with the random tester in all cases. These results demonstrate the effectiveness of the proposed tester.

---

> > > > ### Comment · Reviewer_Si75 · 2022-11-18
> > > > **Reponse to Authors**
> > > >
> > > > Thank you for your timely response and for providing additional evidence. This convinces me of the utility of the approach, in particular in terms of the effectiveness of the tester. Given the high quality and trustfulness of the experiments, the detailed analysis and the engagement of the authors, I will raise my score.

---

### Decision · Program_Chairs · 2023-01-20

**Decision:**

Reject

**Justification For Why Not Higher Score:**

See the aforementioned major concern.

**Justification For Why Not Lower Score:**

N/A.

**Metareview: Summary, Strengths And Weaknesses:**

The main contribution of this work lies in proposing a generate-and-test approach to discovering auxiliary tasks in RL.

After reviewing the authors' rebuttal and an active discussion, the reviewers agree that this paper is generally well-written and easy to read. The proposed approach is simple, easy to understand, and performs well in the tested environments.

A major concern raised by all reviewers is the lack of demonstration of scalability of the proposed approach beyond the current simple environments to more complex ones. To some reviewers, such a concern remains after reading the authors' response.

The authors are encouraged to revise their work based on the reviewers' comments.

**Summary Of Ac-Reviewer Meeting:**

There is sufficient **written** discussion generated on the OpenReview discussion forum to the extent of being able to reach a consensus on the recommendation. Hence, there is no need for a meeting.